# Extreme rejuvenation and softening in a bulk metallic glass

J. Pan [1], Y.X. Wang [1], Q. Guo [2], D. Zhang [2], A.L. Greer [3] & Y. Li [1]

Rejuvenation of metallic glasses, bringing them to higher-energy states, is of interest in improving their plasticity. The mechanisms of rejuvenation are poorly understood, and its limits remain unexplored. We use constrained loading in compression to impose substantial plastic flow on a zirconium-based bulk metallic glass. The maximum measured effects are that the hardness of the glass decreases by 36%, and its excess enthalpy (above the relaxed state) increases to 41% of the enthalpy of melting. Comparably high degrees of rejuvenation have been reported only on microscopic scales at the centre of shear bands confined to low volume fractions. This extreme rejuvenation of a bulk glass gives a state equivalent to that obtainable by quenching the liquid at ~$10^{10}$ K s$^{-1}$, many orders of magnitude faster than is possible for bulk specimens. The contrast with earlier results showing relaxation in similar tests under tension emphasizes the importance of hydrostatic stress.

[1] Shenyang National Laboratory for Materials Science, Institute of Metal Research, Chinese Academy of Sciences, Shenyang 110016, China. [2] State Key Lab of Metal Matrix Composites, Shanghai Jiao Tong University, 800 Dongchuan Road, Shanghai 200240, China. [3] Department of Materials Science & Metallurgy, University of Cambridge, 27 Charles Babbage Road, Cambridge CB3 0FS, UK. Correspondence and requests for materials should be addressed to A.L.G. (email: alg13@cam.ac.uk) or to Y. L. (email: liyi@imr.ac.cn)

A glass is formed on cooling a liquid (if crystallization can be avoided), faster cooling giving more disordered, higher-energy states[1]. The difference between the highest and lowest energies attainable in the glass at a given temperature is remarkable, nearly as large as the enthalpy of melting[2]. After casting, annealing allows relaxation or ageing of glassy states to lower energies, while the opposite process, rejuvenation, can be induced by reheating and faster cooling[3], and, most commonly, by plastic deformation[1, 4]. Deformation broadens the range of interatomic distances in a metallic glass, a clear sign of disordering opposite to the effects of relaxation[5]. Such studies have links with the interest, for crystalline metals, in tailoring properties by control of defect structures at a fixed composition[6].

Viscous flow of metallic glasses near their glass-transition temperature $T_g$ is homogeneous, but their plastic flow at room temperature (RT) shows an instability in which shear is sharply localized in bands that may be as thin as 10–20 nm[7, 8]. This inhomogeneous deformation leads to essentially zero tensile ductility, and is the main impediment to wider structural use of metallic glasses. Rejuvenation reduces the initial yield stress and, it is speculated, could ultimately eliminate the undesirable shear-banding[1]. Extreme rejuvenation is also of interest in exploring the limits of glass formation and stability.

The usual inhomogeneous nature of plastic flow in metallic glasses itself limits the degree of rejuvenation that can be achieved, because the regions of significant strain occupy only a small volume fraction of the specimen. Studies of a single shear band[4, 9] show that the effects of shear can extend into the glassy matrix by some tens of micrometres, far beyond the thickness of the band itself. However, the effects (softening and increased enthalpy) are sharply peaked at the band centre, and the volume fraction of the glass that is strongly affected is small. Rejuvenation in the bands themselves is inefficient as the structural effects of deformation are likely to saturate for shear strains greater than one. Also, rejuvenated states may not be fully retained because of relaxation facilitated by local heating[8].

To achieve significant flow and rejuvenation throughout a deformed metallic glass, it would be helpful for shear-banding to be suppressed. Here we show that under constraint a metallic glass can be compressed to large strains (up to 40%) in a regime of presumed homogeneous flow. The constraint is achieved in notched specimens. Previous work has shown that plastic flow in notched specimens under tension can lead to relaxation rather than rejuvenation[10], and we analyse the distinction between these cases. Under compression, significant volumes of the metallic glass can attain degrees of rejuvenation previously associated only with the central plane of shear bands. The states attained can have energies so high that they would be characteristic of a glassy state obtained by quenching at ~$10^{10}$ K s$^{-1}$.

## Results

**Stress–strain behaviour.** A compressive load was applied at RT to cylindrical bulk-metallic-glass (BMG) specimens with a circumferential notch. The material in the notch region flows under triaxial constraint. Figure 1a shows a specimen with a reduction in the width of the notch by 20% after plastic flow. This apparent 20% compressive axial plastic strain in the disc defined by the notch is accompanied by an increase in the diameter of the disc of only 5.7%. The apparent volume change of the disc (~9% reduction) shows that flow cannot be confined to the disc itself, but must extend into a larger region around the notch. Only a few shear bands with horizontal traces can be observed on the surface of the notch (Fig. 1a). In contrast, the un-notched specimen at similar strain shows (Fig. 1b) many primary shear bands, as expected[8] at ~45° to the loading axis.

The stress–strain curves (Fig. 1c) show the un-notched specimen yielding through shear banding at 1.70 GPa, a stress that stays almost the same with increasing plastic strain. In the notched specimen, lateral constraint of the material in the notch region, and the consequent triaxiality of the stress state (Supplementary Fig. 1), increase the axial stress needed to initiate flow. The material in the notch region is compressed elastically beyond the usual 2.67% strain limit for metallic glasses[11] and yields at 2.30 GPa (i.e. the stress acting on the cross-section of the disc). As deformation proceeds, the disc of material undergoing flow becomes thinner. The resulting increased constraint should cause the effective flow stress to increase. The axial stress does rise to a maximum of 2.64 GPa, but after a true strain of ~19% (derived from the notch width) there is evident softening: at a true strain of 57%, the axial stress has decreased to 2.19 GPa. This decreasing axial stress, despite increasing lateral constraint, can be explained only by a reduction in the actual flow stress of the glass, consistent with ongoing rejuvenation.

**Decreases in microhardness.** Deformed specimens (confirmed to remain fully glassy, Supplementary Fig. 2) were sectioned along the central longitudinal plane to permit mapping of microhardness on the cross-sections. The average Vickers microhardness ($Hv$, kgf mm$^{-2}$) of as-cast specimens is 495 ± 5. In the un-notched specimen compressed to 40% plastic strain, the average $Hv$ is 466 ± 8, a decrease of 5.9% and within the 5–10% range reported[12–17] for deformation by a variety of techniques.

For the notched specimens, we first consider the effects of increasing plastic flow within the disc defined by the notch. Increasing plastic strain (derived from the notch width) reduces the hardness measured on the cross-section along the cylinder axis (Fig. 2a). Compressive plastic strains of 20 and 40% give a uniform $Hv$ of 460 ± 12 and 401 ± 14 respectively along this line, decreases of 6 and 19%. Across the disc diameter (Fig. 2b) there is a similar progressive decrease in $Hv$ as a result of increasing plastic strain. $Hv$ is roughly constant near the centre, suggesting a uniform degree of deformation, but decreases as the periphery of the disc is approached (i.e. at the notch root). In the specimen strained to 40%, $Hv$ at the notch root is only 315 ± 16, a remarkable decrease of 36%.

The specimen strained to 40% was annealed for 12 h at 573 K (~0.9 $T_g$). After annealing, the average $Hv$ is 510 ± 5 (Fig. 2a, b), slightly higher than that of the as-cast specimen, because of structural relaxation. The recovery of $Hv$ in the disc by heat treatment indicates that the extreme softening is caused by structural change in the glass during deformation rather than by extrinsic features such as micro/nano voids or cracks. Cycling between relaxed and rejuvenated states is possible. A fully relaxed specimen ($Hv = 510 ± 5$) when strained to 40% shows $Hv = 336 ± 15$ at the notch root, a decrease of 34% showing that rejuvenation can be repeated. Further annealing restores the fully relaxed state.

A mapping of hardness beyond the immediate region of the notch was undertaken only for a specimen subjected to a compressive plastic strain of 40% (Fig. 2c); this shows that the effects of deformation extend some 2 mm above and below the notch. Noting the axial symmetry, the affected zone has a roughly cylindrical shape. It is of interest to compare the extent of softening in notched and un-notched specimens. The $Hv$ of an un-notched specimen compressed to 40% axial plastic strain would lie on a contour in the centre of the yellow band in the map of the notched specimen taken to 40% nominal strain in the disc (Fig. 2c). The region enclosed by this contour is roughly a cylinder, 2.5 mm in diameter and 3.6 mm in height. Taking this volume in the notched specimen, comparing it with the entire specimen volume when un-notched, and considering the

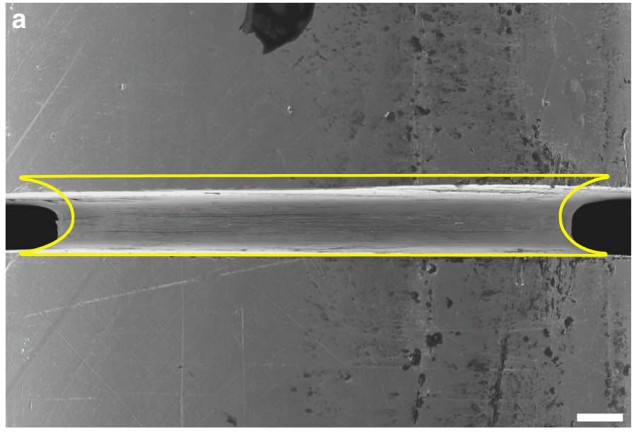

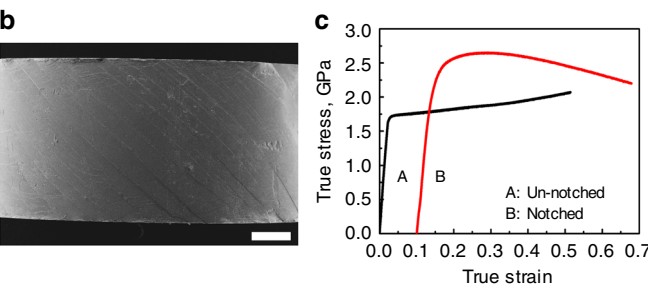

**Fig. 1** Homogeneous flow of a bulk metallic glass by suppression of shear-banding. Cylindrical (4 mm diam.) specimens of a zirconium-based metallic glass are compressed along the cylinder axis (vertical in the scanning electron micrographs in **a** and **b**). **a** The central region of a specimen with a circumferential notch, compressed to 20% axial strain in the disc defined by the notch. The yellow outline shows the shape of the disc before compression. **b** An un-notched specimen compressed to 20% axial strain; the surface shows the traces of multiple shear bands. **c** The true stress–true strain curves of notched and un-notched specimens (curve B is displaced laterally for clarity). The scale bars in **a** and **b** are 200 and 500 μm respectively

mechanical work done in each case, the energy expended per unit volume is almost the same in each case (~3% less in the notched specimen). Within this cylinder in the notched specimen, however, most of the material is softened by much more than the 5.9%, which is characteristic of the deformed un-notched specimen. Thus the constrained geometry in the notched specimen not only allows significantly greater local softening, but also a greater efficiency in achieving overall softening.

Surveying published results, it is clear that under uniaxial compression (i.e. in un-notched specimens), metallic glasses undergo inhomogeneous flow and show (Fig. 3) a consistent trend to lower $Hv$ with increasing plastic strain[4, 14, 17]. For a plastic strain of 40%, the decrease in $Hv$ is ~6%. Shot-peening gives high cumulative plastic strain, but the softening saturates with $Hv$ decreased by ~10%[16]. The $Hv$ decrease in the present un-notched specimen matches this 'inhomogeneous softening' trend.

At the centre of the notched specimens, the decrease in $Hv$ follows a separate, steeper trend: for 40% strain the decrease is 19%. Away from the centre, at the notch root, the decrease of 36% (Fig. 2b) sets a lower plateau. The distinct trends (Fig. 3) show that the flow in the notched specimens has stronger effects on glass structure and properties than the usual inhomogeneous flow in unconstrained uniformly loaded specimens.

**Increases in heat of relaxation**. In notched specimens, the non-uniform loading and flow may lead to significant residual stresses

after deformation, and in metallic glasses, it is known that these can affect measured values of $Hv$[16]. The extent of rejuvenation may therefore be more reliably quantified as the exothermic heat of relaxation $\Delta H_{rel}$ on heating up to $T_g$[1, 2, 18] (Fig. 4a). Deformed notched specimens were sectioned to retain just the disc within the notch region (Supplementary Fig. 3), portions of which were characterized. For comparison, a deformed un-notched specimen was also studied. The $\Delta H_{rel}$ of 0.49 kJ mol$^{-1}$ in the as-cast glass rises by 33% to 0.65 kJ mol$^{-1}$ in the un-notched specimen strained to 40% (Fig. 4b). For the whole disc in a notched specimen strained to 40%, $\Delta H_{rel}$ increases by 131% to 1.13 kJ mol$^{-1}$. When the edge of the disc is selected (i.e. at the notch root), $\Delta H_{rel}$ is 3.42 kJ mol$^{-1}$. The deformed specimens when annealed (12 h at 593 K, as before) show $\Delta H_{rel} = 0.09$ kJ mol$^{-1}$ (Fig. 4b), lower than that of the as-cast glass and consistent with the recovery in $Hv$ on annealing (Fig. 2a, b).

The higher values of $\Delta H_{rel}$ found in the deformed disc of a notched specimen, and particularly at its periphery, correspond well with the lower values of $Hv$ (Fig. 2b), and confirm that extreme rejuvenation can be achieved.

**Effect of temperature**. Plastic deformation damages the glass and the rise in energy is opposed by relaxation. The achievable degree of rejuvenation is determined by the balance of damage and relaxation rates[1]. Lower temperature impedes relaxation and thereby facilitates rejuvenation[1, 19]. For comparison with the results at RT, a notched specimen was compressed at 77 K to a local strain of 33% (Supplementary Fig. 4). Measurements on the cross-section show $Hv = 440 \pm 13$ at the centre of the specimen and $335 \pm 17$ at the notch root, where $\Delta H_{rel} = 3.29$ kJ mol$^{-1}$. These values are remarkably similar to those expected for the same strain applied at RT.

**Instrumented indentation**. Instrumented-indentation tests on the section of the notched specimen strained to 40% at RT show that the behaviour depends on the position of the indent relative to the notch. The loading and unloading curves show that the hardness and indentation modulus decrease (ultimately by 31 and 8% respectively) comparing a region far from the notch, with the centre of the disc and with the periphery of the disc (notch root). Importantly, the shape of the indents also changes. Accompanying the decreases in hardness, the pile-up height decreases (Fig. 5b–d, confirmed in microhardness tests, Supplementary Fig. 5). The pile-up is absent for indents at the notch root. The same effect is found on approaching the centre of a shear band[4], and is consistent with decreased work-softening when deforming a metallic glass that is already heavily deformed[20].

Independent of indent location, the loading curves (Fig. 5a) show deviations from the Hertzian shape, suggesting pop-ins related to shear-banding (confirmed by compression tests on micropillars extracted from a deformed specimen, Supplementary Fig. 6).

**Discussion**
In metallic glasses, a shear band is activated when the stress over a complete shear plane exceeds a critical limit[21]. In the present work, the constraints on the flow of the glass in notched specimens (where no external slip steps are possible) is likely to make shear-band operation difficult. The reductions in hardness (Fig. 3) imply two distinct regimes in un-notched and notched specimens. The greater constraint in the latter case leads to greater, not lesser softening. And, as noted earlier, the constrained flow induces greater softening for a given energy input per unit volume, and more of that input remains as stored energy in the metallic glass. We therefore presume, even without direct

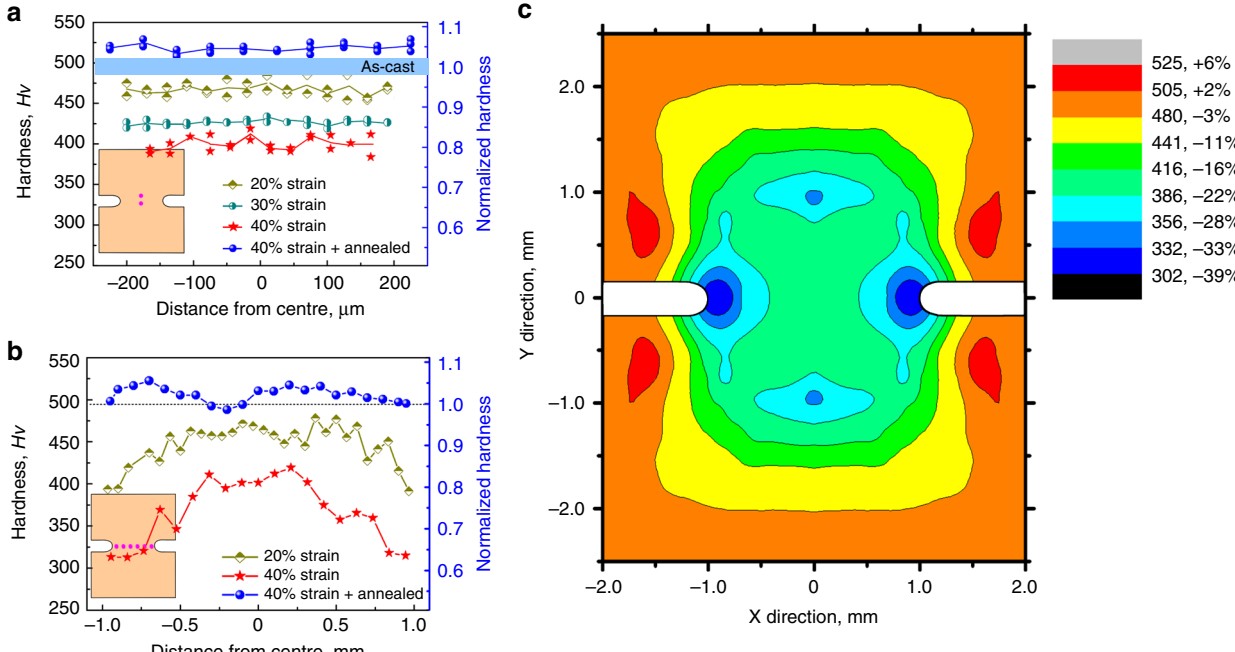

**Fig. 2** Deformation-induced softening in bulk metallic glasses. **a**, **b** Profiles of Vickers microhardness $Hv$ of notched cylinders of $Zr_{64.13}Cu_{15.75}Ni_{10.12}Al_{10}$ glass after compression to the indicated strains and subsequent annealing. As in Fig. 1a, the strains are for the disc defined by the notch. **a** $Hv$ profiles along the central axis of the disc (inset). **b** $Hv$ profiles across the diameter of the disc (inset). **c** Hardness contour map showing the extent of softening after 40% strain

microstructural evidence, that the flow in the notched specimens represents a different regime and is predominantly homogeneous. This assumption is strongly supported by the correspondence between diminishing indent pile-ups (Fig. 5) and softening . This correlation can be seen both in the present results on notched specimens and, spatially resolved, in the matrix close to a single shear band in a BMG specimen[4]. In the latter case, the softening occurs in material that is likely to have undergone some homogeneous deformation and that shows no local shear bands.

Increases in $\Delta H_{rel}$ after plastic deformation can be considered as stored energy of cold work[22]. For the un-notched specimen subjected to 40% plastic strain, the increase in $\Delta H_{rel}$ is 0.16 kJ mol$^{-1}$, and the mechanical work done (WD) calculated from the stress–strain curve (Fig. 1c) is 10.4 kJ mol$^{-1}$: the stored energy is ~1.5% of the WD, a fraction similar to that for polycrystalline metals[23] and to published values for metallic glasses[22]. For the notched specimen subjected to the same strain in the disc, the energies are first calculated assuming that the only relevant volume is that of the disc defined by the notch. The increase in $\Delta H_{rel}$ is overall 0.64 kJ mol$^{-1}$ for the disc and the WD is 14.2 kJ mol$^{-1}$: these values give the stored energy as ~4.5% of the WD. From Fig. 2c, however, it is clear that rejuvenation, and therefore stored energy, are not confined to the disc. The volume of material expected to be affected similarly to that at the centre of specimen lies in a cylinder roughly 2 mm in diameter and 2.6 mm in height. The total stored energy that this larger volume implies would amount to ~29% of the WD. This is a much higher fraction than previously reported for deformed metallic glasses, and supports the point made above, in considering $Hv$ values, that deformation under constraint is particularly efficient in inducing rejuvenation. Even so, the fraction of WD that is stored is still lower than some values for polymeric glasses[24]. And rejuvenation by constrained plastic flow is inefficient in generating stored energy, when compared with initial elastostatic loading, in which, however, storage saturates at very low strains[2].

The maximum $\Delta H_{rel}$ in the present work, 3.42 kJ mol$^{-1}$, is 66% higher than the highest previously reported, which was in a specimen subjected to 50 revolutions (maximum strain ≈ 37) in high-pressure torsion[25]. This maximum enthalpy excess of the deformed glass over the relaxed glass (i.e. the glass after release of $\Delta H_{rel}$) is 41% of the enthalpy of melting. For comparison, the heats of crystallization of metallic glasses are ~40% of the enthalpy of melting[1]. Thus the difference in enthalpy between possible glassy states is essentially the same as that between the relaxed glass and the state to which it would crystallize.

The enthalpy of an as-cast glass is dependent on the cooling rate at which it was formed. The glass transition occurs at critical value of liquid viscosity $\eta$ that is inversely proportional to the cooling rate. At the standard cooling rate of 20 K min$^{-1}$, the critical $\eta$ is taken to be $10^{12}$ Pa s. From the measured temperature dependence of $\eta$ for Zr-based glass-forming liquids[26], the effective value of the glass-transition temperature $T_g$ can then be estimated for other cooling rates. Given the dependence of $T_g$ on cooling rate, the excess enthalpy (relative to that of a glass formed at the standard cooling rate) can be estimated from the temperature dependence of the liquid enthalpy[27], as shown in Fig. 1 of ref. [2]. In this way, the excess enthalpy ($\Delta H_{rel}$) of a glass can be related to the cooling rate at which a glass of that enthalpy would have formed from the liquid without further processing.

Using this approach, a general correlation has been proposed between $\Delta H_{rel}$ and effective cooling rate[1]. From this correlation, the highest stored energy in the present work corresponds to cooling at nearly $10^{10}$ K s$^{-1}$ (Fig. 6). Such a rate may be achieved, for example, by laser surface melting[28], but is 7–8 orders of magnitude higher than could be realized in cooling of a BMG. Thus the attraction of mechanical processing becomes clear: a bulk specimen can be brought into a state characteristic (at least in terms of its overall energy) of a much faster quenched glass. Not only is such a state expected to have better mechanical properties, but its bulk permits more comprehensive

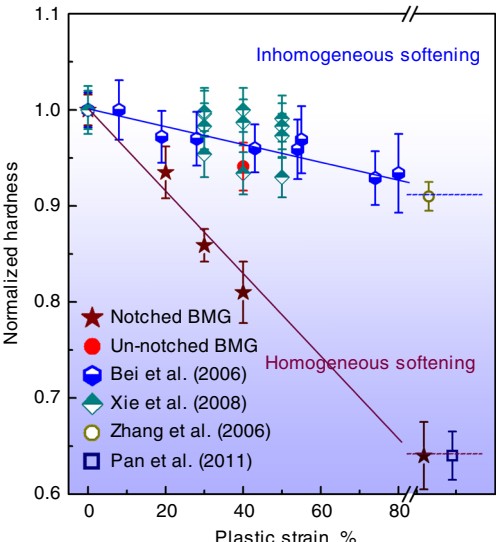

**Fig. 3** Published data[4, 14, 16, 17] on the hardness values of BMGs, normalized with respect to the hardness of as-cast (undeformed) specimens. Deformation by uniaxial compression of un-notched specimens, indentation or shot-peening causes inhomogeneous flow mediated by shear bands. The normalized hardness decreases roughly linearly with applied strain. Compression of notched specimens in the present work gives homogeneous flow in the notch region and much greater decrease in hardness. The dashed lines to the right indicate possible limiting degrees of softening achieved at large (unspecified) strains by shot-peening[16], at the heart of a shear band[4], and at the notch root (in the present work). Error bars are the standard deviation for the measurements

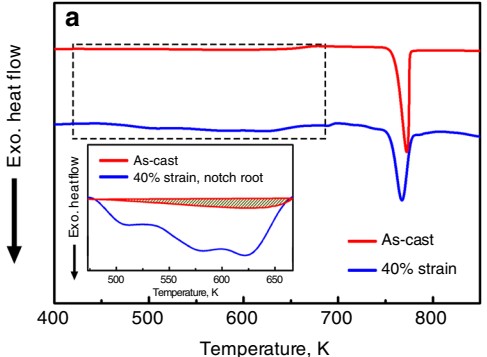

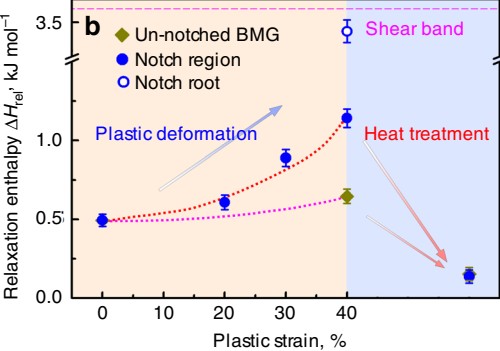

**Fig. 4** Extreme rejuvenation by compression of notched glass specimens. **a** Differential scanning calorimetry traces (heating rate 20 K min⁻¹) of $Zr_{64.13}Cu_{15.75}Ni_{10.12}Al_{10}$ glass: an as-cast specimen and a notched specimen compressed to 40% nominal strain. **b** Relaxation enthalpy $\Delta H_{rel}$ of specimens of this glass compressed to different strains and annealed. The $\Delta H_{rel}$ value at the heart of a shear band[4] is also given. Error bars are the standard deviation for the measurements

characterization of structure and properties. A surface layer quenched at $10^{10}$ K s⁻¹ would be limited to a thickness of order 1 μm[28]. In contrast, the effectively rejuvenated bulk in the notched specimen is nearly 3 mm thick (Fig. 2c), and that may be far from a fundamental limit. Similar or higher degrees of rejuvenation may be achievable by irradiation[1], but treating bulk specimens is likely to require neutron irradiation[29], for which radioactivation is a problem and there is limited access to processing facilities. As shown in Fig. 6, the degree of rejuvenation achieved by triaxial compression in the present work far exceeds that achieved by other plastic deformation methods, or by different methods such as elastostatic loading[2] or thermal cycling[30].

The present work has focused on rejuvenation yet, as noted above, plastic flow in notched specimens under tension can in contrast induce structural relaxation[10]. In the early work of Spaepen[31] it was recognized that there are competing processes when a metallic glass is subjected to shear. The shear flow induces disordering and thermally activated diffusional rearrangements control the rate of re-ordering. The state of the glass can be characterized in many ways, of which free volume (as chosen by Spaepen) is one of the most common. The reduced free volume per atom is defined by $x = v_f/\gamma v^*$, where $v_f$ is the free volume per atom, $\gamma$ is a geometrical factor (of value 0.5–1.0), and $v^*$ is the critical free volume for an atomic jump. Shear of the metallic glass forces atoms into spaces that are too small, expanding the nearest-neighbour atomic cage, and increasing the volume of the system. Adopting the analyses already used[10, 31], the rate of generation of free volume is related to the applied shear stress $\tau$ by:

$$\frac{dx}{dt} = \frac{f}{\gamma}\exp\left(-\frac{\Delta G^m}{kT}\right)\exp\left(-\frac{1}{x}\right)\left[\cosh\left(\frac{\tau\Omega}{2kT}\right) - 1\right]\frac{2kT}{xSv^*},\quad (1)$$

where $f$ is the atomic vibration (attempt) frequency, $\Delta G^m$ is the activation energy for atomic motion, $\Omega$ is the atomic volume, $S$ is an elastic modulus (defined by $S=2\mu(1+\nu)/3(1-\nu)$, where $\mu$ is shear modulus and $\nu$ the Poisson ratio), and $k$ and $T$ have their usual meanings.

Thermally activated rearrangements enable the reduced free volume to evolve towards the value $x_{eq}$ that represents equilibrium for the ambient conditions. These rearrangements occur at a rate determined not only by the temperature but also by the hydrostatic (mean) component ($\sigma_m = (\sigma_1 + \sigma_2 + \sigma_3)/3$) of the stress. For metallic glasses[32, 33], the activation volume $V$ is in the range $(0.6–1.0)\Omega$, and we take $V = 0.8\Omega$. With this positive value, atomic motion is significantly accelerated under tensile stress and vice versa. The rate of free-volume relaxation is given by[10]:

$$\frac{dx}{dt} = -fx^2\left[\exp\left(-\frac{1}{x}\right) - \exp\left(-\frac{1}{x_{eq}}\right)\right]\exp\left(-\frac{\Delta G^m - \sigma_m V}{kT}\right).\quad (2)$$

With input parameters for $Zr_{64.13}Cu_{15.75}Ni_{10.12}Al_{10}$ metallic glass (Methods and ref. [10]), Eqs. (1) and (2) can be used to estimate the change in free volume during plastic flow, an increase representing rejuvenation or a decrease representing relaxation. For notched specimens, two important effects arise from the triaxial constraint. Firstly, a decreased effective shear stress means that the flow in the notched region is much less than in an un-notched specimen loaded with the same axial stress. Secondly, the hydrostatic component of the stress is somewhat larger.

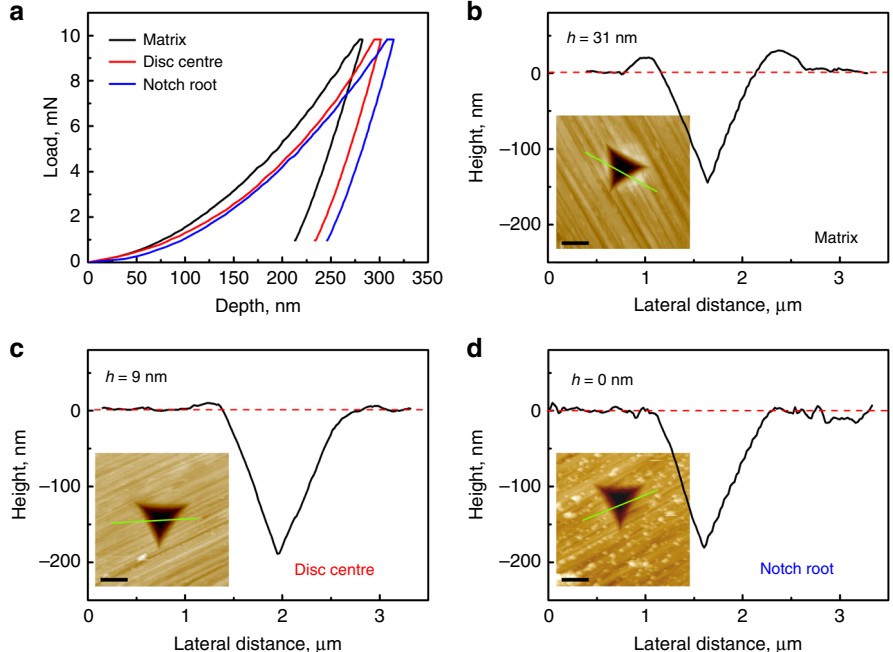

**Fig. 5** Nanoindentation of a deformed glass. A notched specimen was subjected to 40% compressive strain (axial strain in the disc defined by the notch) at RT. **a** loading/unloading curves for indentation of the matrix (on a specimen cross-section away from the notch region), at the centre of the circular face of the deformed disc sectioned from the deformed specimen (Supplementary Fig. 3), and at the edge of the circular face (the notch root). **b–d** For these three locations, AFM line-scans and images of indents show a progressive decrease in the height ($h$) of the pile-up, which disappears in the region of the notch root. The scale bars in the insets of **b–d** are 1 μm

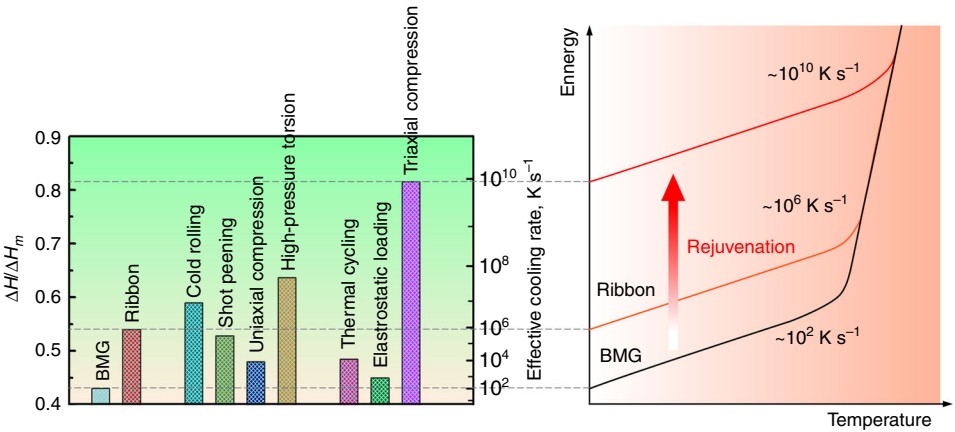

**Fig. 6** Relative enthalpies of metallic glasses (normalized by the heat of melting $\Delta H_m$ and relative to the annealed crystal[1]) in different as-cast conditions, and subjected to plastic deformation, or to other processes such as elastostatic loading[2] and thermal cycling[30]. Higher relative enthalpy corresponds to glassy states expected to be achieved at higher cooling rates

In a notched specimen, at a given axial stress, the flow rate in the central region is the same under tension and compression, and accordingly the rates of free-volume generation (from Eq. (1)) are the same (Fig. 7a). Under the conditions explored in ref. [10], the rate of relaxation ($dx/dt$) under zero stress is $10^{-9}$ s$^{-1}$; this rate is higher/lower for increasing tensile/compressive values of $\sigma_m$ (Fig. 7a). Under tension, the rate of annihilation can exceed the rate of generation so that, as reported previously[10], the free volume can decrease when sufficient stress is applied (Fig. 7b). Under compression, in contrast, the free volume increases.

For an un-notched specimen flowing under uniaxial stress, whether tensile or compressive, the rate of generation of free volume far exceeds its possible rate of decrease, so that

rejuvenation is expected (Fig. 7b and Supplementary Fig. 7). The increase in free volume is indicated to be much faster for this uniform flow, than for flow in a compressed notched specimen. In an actual un-notched specimen, however, uniform flow would not be maintained; there is an instability (analysable in terms of free volume[34]) leading to shear banding. Flow on one dominant shear band would lead to rapid catastrophic failure, precluding significant overall rejuvenation. The key benefit of using a notched specimen is to allow higher stresses to be reached without shear banding. Shear banding can also be avoided in an un-notched specimen by deforming at elevated temperature; in this creep regime, the free volume in the specimen does increase, though it is noted that the generation process is inefficient[35].

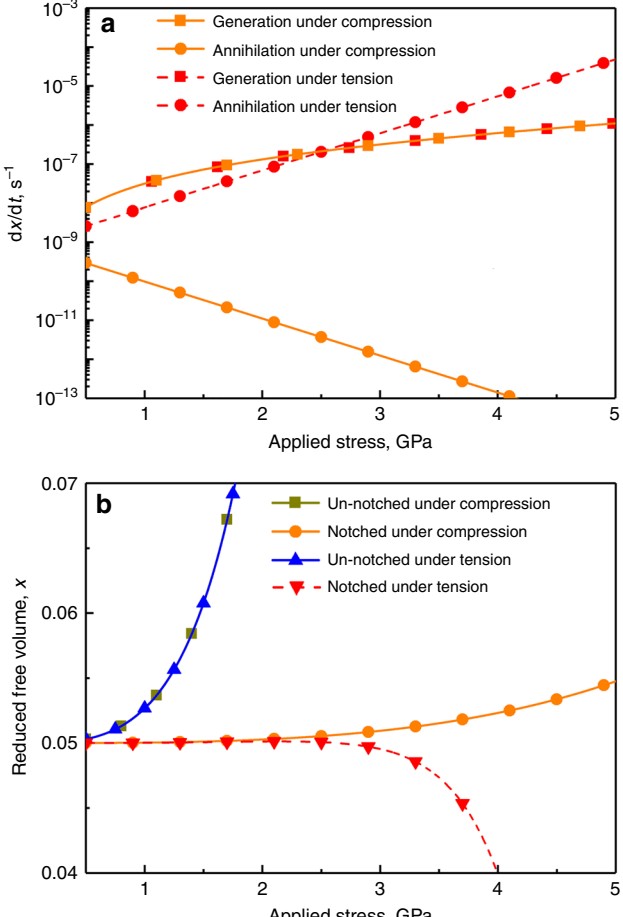

**Fig. 7** Effect of stress state on the evolution of free volume. **a** The generation and annihilation rates of free volume in a notched BMG specimen as a function of applied axial compressive or tensile stress. **b** The evolution of reduced free volume $x$ in a notched BMG specimen as a function of applied compressive or tensile axial stress. The values for an un-notched specimen are included for comparison

The rates shown in Fig. 7a are for loading at RT. At applied compressive stresses sufficient to achieve a significant increase in free volume, the rate of annihilation of free volume is many orders of magnitude lower than the rate of generation. The annihilation rate is thus negligible, explaining why the reduction in the rate by testing at 77 K rather than RT has no effect on the overall rejuvenation. This also suggests that, at least, the central region of a notched specimen is over-constrained. The degree of stress triaxiality could be reduced to permit more flow while still stifling free-volume annihilation; this is of course the condition at the notch root where the greatest rejuvenation is seen.

The good correspondence between reduced $Hv$ and increased $\Delta H_{rel}$ suggests that the reductions in $Hv$ are influenced mainly by the degree of rejuvenation of the specimen rather than by any residual stresses. The map of $Hv$ in Fig. 2c shows (in red) regions where $Hv$ has increased relative to the undeformed material far from the notch. As the specimen is compressed axially, the central region must expand laterally, in effect applying a tensile hoop stress to the annular regions just above and below the notch. It may be that these regions do undergo relaxation, analogous to the effects of tension previously reported[10].

The maximum hardness decrease of 36% in the material at the notch root (Fig. 2c) is the same as the decrease at the centre of a shear band[4]. In the latter case, increasing the shear offset on the

band, the affected zone widened, but the hardness decrease at the centre remained at 36%, suggesting saturation. It remains unclear if a similar saturation effect applies for deformation of the type applied in the present work.

Using a geometric constraint (compression of a circumferentially notched cylinder), we have imposed plastic flow, likely to be largely homogeneous, on a Zr-based BMG up to 40% strain. This treatment combines the efficiency of energy storage associated with homogeneous flow (previously seen for loading in the elastic regime, but only to very small strains) with large strain. The deformed glass shows decreases in hardness and increases in stored energy previously seen only at the heart of shear bands and not in bulk. The constrained loading is much more efficient than simple uniaxial compression in rejuvenating the BMG: for a given work done per unit volume, greater softening is achieved, and a higher fraction of the WD (nearly 30%, roughly one order of magnitude higher than reported before by plastic deformation) remains as stored energy in the metallic glass. The maximum stored energy is 66% greater than any previously achieved by plastic deformation, and roughly four times higher than any achieved by mechanical treatments in the elastic regime. The deformed BMG can reach a state characteristic of cooling at $\sim 10^{10}$ K s$^{-1}$, some eight orders higher than the rate at which it was originally cast. While it improves the plasticity of the glass, this degree of rejuvenation has not yet eliminated the shear-banding instability, a goal that remains for future work. As previous work on the same BMG[10] has shown that constrained flow in tension can induce structural relaxation, the present results, showing rejuvenation under compression, further emphasize the importance of the hydrostatic stress component in understanding the effects of plastic flow on the structure of metallic glasses. These effects may be most apparent when the flow is homogeneous, as is presumed to be largely the case in the notched specimens studied in ref. [10] and in the present work. Comparison of the effects of deformation at RT and at 77 K suggests that, under compressive, relaxation processes in the BMG are effectively stifled; the observed rejuvenation can then be understood to involve only damage (i.e., in one description, the generation of free volume) in the BMG. Mapping of the extent of rejuvenation over a cross-section through a notched sample reveals a complex pattern. Further analysis of this behaviour should assist in optimizing methods for bulk rejuvenation of metallic glasses.

## Methods

**Specimen fabrication.** A BMG with a nominal composition of $Zr_{64.13}Cu_{15.75}$-$Ni_{10.12}Al_{10}$ (at.%) was prepared by arc-melting mixtures of high-purity metals (above 99.9%) in a titanium-gettered high-purity argon atmosphere. Cylindrical specimens with a diameter of 4 mm and a length of 75 mm were fabricated by tilt-casting into a copper mould. The structure of the cast specimens was characterized and confirmed to be fully glassy by X-ray diffraction (XRD) employing a Bruker AXS (D8 ADVANCE) instrument with CuKα radiation at 40 kV, and by transmission electron microscopy (JEM, 2010F) with a field-emission gun. Some of the cylinders were notched circumferentially by gently grinding in a custom-made machine, followed by fine polishing and final cleaning in an ultrasonic bath. The notches had a width of 400 μm and a depth of 1 mm.

**Mechanical processing.** Compressive loadings were carried out on a specimen of 8 mm in height at RT or at 77 K using an Instron 5982 machine with a maximum load of 100 kN at a cross-head speed of 0.05 mm min$^{-1}$. The specimens subjected to different deformations are listed in Supplementary Table 1. The stress and strain in notched specimens are calculated based, respectively, on the circular area of the disc and on its height defined by the notch. For comparison, un-notched cylindrical specimens with an aspect ratio (height:diameter) of 0.6 were also studied. The surface morphologies of the specimens before and after deformation were examined by scanning electron microscopy (FEI Quanta 600).

**Specimen characterization.** To characterize the effects of deformation, micro-hardness tests were conducted on specimen cross-sections using a Vickers diamond indenter with a load of 50 g and a dwell time of 10 s. Notched specimens deformed to 40% compressive strain were annealed at 573 K for 12 h to reveal the effects of

structural relaxation; the still-glassy structure of these specimens was confirmed by XRD as above. The hardness map (Fig. 2c) involved making some 2000 indents. As the sample has obvious vertical and horizontal mirror plane symmetries, the raw data were averaged to give the same pattern in each quadrant.

Instrumented indentation (Agilent Nano Indenter G200) was with a Berkovich diamond tip, applying a maximum load of 10 mN with a loading rate of 0.5 mN s$^{-1}$. The hardness in different regions was measured by the method of Oliver and Pharr[36]. The indent morphology was subsequently observed using atomic force microscopy (AFM, Bruker MultiMode 8).

The thermal response of as-cast and deformed specimens was investigated with differential scanning calorimetry (DSC, TA Q2000) at a heating rate of 20 K min$^{-1}$ in a flow of argon. After first heating up to 873 K (completely crystallizing the specimen), a second run under identical conditions was used to determine the baseline for each measurement. For calculation of the relative relaxation enthalpy, the melting enthalpy of the Zr-based glass in the present work was measured as 8.31 kJ mol$^{-1}$. The melting enthalpy was measured by simultaneous thermal analyser (Netzsch STA 449 F3) by heating at 20 K min$^{-1}$ up to 1473 K.

**Free-volume calculations**. To account for the effects of hydrostatic stress, we used a modified free-volume model to estimate the structural changes during deformation[10]. For an un-notched specimen, the shear stress (according to the von Mises yield criterion) is given by $\tau = \sigma/\sqrt{3}$, where $\sigma$ is the applied axial stress, and the hydrostatic (mean) stress is given by $\sigma_m = \sigma/3$. For a notched specimen, the stresses are calculated using the analytical treatment of Neuber[37]. Approximating the actual notch shapes, we derive that at the centre of the specimen the principal stresses are $0.59\sigma$ axially, and $0.36\sigma$ biaxially in the cross-sectional plane; with these values, $\tau = 0.13\sigma$, and $\sigma_m = 0.44\sigma$ (Supplementary Fig. 1). For simplicity, these conditions at the centre of the specimen are used to calculate the free-volume generation and annihilation rates (Eq. (1) and Eq. (2)), as the conditions apply approximately throughout a significant volume (Fig. 2c) in the notched region. The values of parameters used in the calculation are: $T = 300$ K, $\gamma = 0.15$, $f = 5.415\times10^{12}$ s$^{-1}$, $\Omega = 2.424\times10^{-29}$ m$^3$, $\mu = 28.5$ GPa, $v^* = 1.939\times10^{-29}$ m$^3$, $\Delta G^m = 10^{-19}$ J, $x = 0.05$, and $\nu = 0.377$[38, 39].

**Data availability**. The data acquired in the course of this study are available from corresponding author Y. Li on request.

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

## Acknowledgements

The authors thank Prof. H.J. Gao for useful discussions. J.P. thanks Mr. J. Zhang for assistance with hardness measurements. J.P. and Y.L. acknowledge support from the National Natural Science Foundation of China under Grant nos 51401220 and 51471165, and A.L.G. from the European Research Council under the European Union's Horizon 2020 research and innovation programme (grant ERC-2015-AdG-695487: ExtendGlass).

## Author contributions

Y.L. and A.L.G. designed and supervised the project. J.P. synthesized the notched specimens, and performed the compression testing, hardness and DSC experiments. Y.X.W. carried out the nanoindentation and AFM testing. Q.G. and D.Z. fabricated the micro-pillars and carried out the micro-compression. J.P., Y.L. and A.L.G. analysed the data. Y. L. and A.L.G. wrote the paper. All authors discussed the results and commented on the manuscript.
