## [Peer Review File · Nature Communications]

Reviewers' comments:

Reviewer #1 (Remarks to the Author):

This is a very interesting paper which merits publication in Nature Communication. In this article the authors describe the mechanical properties of a metallic glass strongly rejuvenated by notched compression treatment at room temperature. The results show that the portion of the sample near the notch was extremely rejuvenated to the state equivalent to the sample obtained by the cooling rate of 10^{10} K/sec. The sample is softer, stronger and more ductile. Because of the groove around the cylinder mechanical deformation does not spread, but is spatially confined. This is a really nice work, but I have a few comments.

They claim that the deformation is homogeneous, but there is no direct evidence of spatial homogeneity at a nm level. It may well be that the sample simply has a very high density of shear bands. The TEM picture (Fig. S2) confirms the absence of crystallinity, but says little about the shear bands. They should either present direct evidence, or soften the claim of spatial homogeneity.

Also the deformed area must extend substantially above and below the notched region, probably much more than suggested by Figure S1 (b). Because the notched area is reduced in volume (substantial decrease in height without increase in diameter), deformation must be reaching deep into the cylinder, possibly mms. In Figure 2 (a) if they extended the measurement beyond 200 μm they would have found the extent of the deformed portion. It is possible that deformation extends deeper near the notch rather than at the center. It will be useful, for application as well, to know how deep the deformed area is and how much volume was rejuvenated.

Reviewer #2 (Remarks to the Author):

The authors present a study on mechanical rejuvenation in metallic glasses. The shown rejuvenation was achieved by a notched specimen geometry under compressive load.

The paper is well written and a large number of experiments have been conducted. However, it is a mystery to me how the authors come to the opposite conclusions than in their previously published work in Physical Review Letters. Even though several authors of the current study are also authors on the paper in PRL, they do not even cite or discuss their earlier findings. This is somewhat shocking, especially because the authors conduct the same study but find the opposite. I am tempted to conclude that the authors do not know what they are actually investigating, or they just aim at publishing whatever they find/believe they find.

Clearly, this work cannot be published.

In addition to this very sketchy situation, there are numerous issues with the manuscript and the presented findings. In fact, the topic of mechanically driven rejuvenation is much more complex than suggested by the current manuscript, and the authors fail in adequately discussing those. Since the complexity of the topic is high, as the authors certainly are aware of, the results belong to a more specialized journal. This requires showing that the data is true and why it is different to the earlier work.

Reviewer #3 (Remarks to the Author):

This work reports rejuvenation and softening of metallic glass, which has been recently attracted much attention in the field of metallic glasses due to scientific and practical reasons. Un-notched and notched specimens were loaded in compression at low and room temperatures. The authors observed

extreme rejuvenation and softening in the notched specimen. Achieved degree of rejuvenation is a high level, which cannot be realized by the conventional quenching rate. The manuscript is well written. The methodology is reasonable. The result is interesting and a current focus of the metallic glass community. The discussion is basically supported by various experiments, such as TEM imaging, DSC, micro-indent tests, nanoindentation tests of pillars. On the other hands, I have some concerns and comments as below. Before recommending the manuscript for publication in nature communications, I suggest to address them.

[1] Page 1, line 22

"We explore the effects of this rejuvenation, possibly near the achievable limit, on mechanical properties."

In this work, the degree of rejuvenation is defined by the energy increase or the hardness decrease from the as-cast state. Therefore, the degree of rejuvenation depends on the initial state. Is it possible to discuss the achievable limit of the rejuvenation based on this definition? If we prepare a well-aged glass and take it as an initial state, the upper limit of the rejuvenation becomes higher when we apply the present definition.

[2] Page 4, line 147

"the enthalpy excess of the deformed glass over the relaxed glass is essentially the same as that of the relaxed glass over the state to which it would crystallize."

Does "the relaxed glass" mean the as-cast glass? As indicated in the previous comment [1], excess enthalpy of the deformed glass over the relaxed glass depends on the energy state of "the relaxed glass". Therefore, there is a possibility that this sentence does not work well.

[3] Page 4, line 150

"Scaling calorimetric and viscosity data typical for a Zr-based BMG_{1,3,22,23}, the highest energy in the present work corresponds to cooling at nearly 10^{10} K/s (Fig. 3c)."

How do you estimate the cooling rate of 10^{10} K/s from the energy? In other words, how do you scale calorimetric measurement and viscosity data. Since " 10^{10} K/s" is one of keywords in this manuscript, it is better to provide a further explanation about this estimation in the "Methods" or in the supplementary information.

[4] Page 5, line 170

"That these values are similar to those expected for the same strain applied at RT suggests that the rejuvenation is reaching saturation."

In the case of 77 K, the maximum strain is 33%, which is smaller than 40 % of RT. Thus, there still remains a possibility that the degree of rejuvenation becomes higher when we applied a larger strain than 33% at 77K. Compressive loading with a strain larger than 40 % at RT or than 33% at 77K should demonstrate the saturation more clearly and directly. The saturation is an important point of this work.

Comments by Reviewer #1

We thank the reviewer for recognizing the key message of the paper.

They claim that the deformation is homogeneous, but there is no direct evidence of spatial homogeneity at a nm level. It may well be that the sample simply has a very high density of shear bands. The TEM picture (Fig. S2) confirms the absence of crystallinity, but says little about the shear bands. They should either present direct evidence, or soften the claim of spatial homogeneity.

We accept the reviewer's point that we have no direct evidence at a nm level, but we consider that there is significant indirect evidence that the deformation is homogeneous. We now discuss this point at the beginning of the Discussion section (pages 6 to 7); the wording is intended to soften (but still maintain) our claim that the deformation is homogeneous:

“In metallic glasses, a shear band is activated when the stress over a complete shear plane exceeds a critical limit (Ref. [21]). In the present work, the constraints on the flow of the glass in notched specimens (where no external slip steps are possible) is likely to make shear-band operation difficult. The reductions in hardness (Figs. 3) imply two distinct regimes in un-notched and notched samples. The greater constraint in the latter case leads to greater, not lesser softening. And, as noted earlier, the constrained flow induces greater softening for a given energy input per unit volume, and more of that input remains as stored energy in the metallic glass. We therefore presume, even without direct microstructural evidence, that the flow in the notched samples represents a different regime and is predominantly homogeneous. This assumption is strongly supported by the correspondence between diminishing indent pile-ups (Fig. 5) and softening (i) in the present notched specimens and (ii) in the matrix close to a single shear band in a BMG sample [Ref 4]. In the latter case, the softening occurs in material that is likely to have undergone some homogeneous deformation and that shows no local shear bands.”

Also the deformed area must extend substantially above and below the notched region, probably much more than suggested by Figure S1(b). Because the notched area is reduced in volume (substantial decrease in height without increase in diameter),

deformation must be reaching deep into the cylinder; possibly mms. In Figure 2(a) if they extended the measurement beyond 200 μm they would have found the extent of the deformed portion. It is possible that deformation extends deeper near the notch rather than at the center. It will be useful, for application as well, to know how deep the deformed area is and how much volume was rejuvenated.

This is an excellent suggestion, and we thank the reviewer for raising this point. Following the suggestion, we have made new measurements to map the hardness distribution more than 2 mm above and below the notched region. As suggested by the reviewer, the deformation-induced softening indeed extends far away from the notch, occupying a cylindrical volume some 2.5 mm in diameter and 3.6 mm in height. This volume is much larger than we had thought, which of course does have positive implications for the usefulness of this rejuvenation technique. Within this cylindrical volume in the notched specimen, most of the material is softened by much more than the 5.9% characteristic of the deformed un-notched specimen. These observations verify the extreme softening and rejuvenation, and show that the effects extend to a larger volume. In the revised manuscript, we have added Fig. 2(c) to show the contour map of the hardness, and there is additional text to accompany this (page 4). Thanks to the reviewer's comment, and to our new mapping work in response, the manuscript is, we believe, now much improved.

Comments by Reviewer #2

The authors present a study on mechanical rejuvenation in metallic glasses. The shown rejuvenation was achieved by a notched specimen geometry under compressive load. The paper is well written and a large number of experiments have been conducted. However, it is a mystery to me how the authors come to the opposite conclusions than in their previously published work in Physical Review Letters. Even though several authors of the current study are also authors on the paper in PRL, they do not even cite or discuss their earlier findings. This is somewhat shocking, especially because the authors conduct the same study but find the opposite. I am tempted to conclude that the authors do not know what they are actually investigating, or they just aim at publishing whatever they find/believe they find. Clearly, this work cannot be published.

We really thank the reviewer for this comment, because it highlights a significant deficiency in the original presentation of our results. We must apologize for our original text which led to confusion and misunderstanding. The present results on rejuvenation are, in fact, entirely consistent with the earlier work [Wang et al., *PRL* **111** (2013) 135504] on notched specimens showing relaxation, the opposing effects arising from the difference between compressive and tensile loading. This difference was already foreseen in the closing comments about compressive loading in Wang et al. (2013). The original manuscript was prepared with a shorter word-limit in mind and with a focus on the newly found dramatic rejuvenation effects; those factors give rise to our omitting to cite Wang et al. (2013).

We sympathize with the reviewer's comment that '*the authors conduct the same study but find the opposite*', but (perhaps surprisingly) there are significant differences between tensile and compressive loading, so the present work (in compression) is very much not the same as the earlier study [Wang et al. (2013)] (in tension). In the revised manuscript, we cite the earlier study as Ref. [10]. Due to the effect of the hydrostatic component of the applied stress, plastic flow in the notched samples under tension can cause relaxation, while under compression it leads to rejuvenation. A compressive hydrostatic stress retards the annihilation of free volume, while the effect of tensile hydrostatic stress is opposite. To clarify this distinction, we have added to the text (pages 8 to 10), and a new figure (Fig. 7). And we also have appended the Supplementary Information, in particular Fig. S7.

We hope that the revised manuscript (including Supplementary Information) addresses this particular concern of the reviewer.

In addition to this very sketchy situation, there are numerous issues with the manuscript and the presented findings. In fact, the topic of mechanically driven rejuvenation is much more complex than suggested by the current manuscript, and the authors fail in adequately discussing those. Since the complexity of the topic is high, as the authors certainly are aware of, the results belong to a more specialized journal. This requires showing that the data is true and why it is different to the earlier work.

We certainly agree with the reviewer that the topic of rejuvenation is complex. The state of a glass cannot be fully described by a single parameter. However, we consider that the reduced hardness and raised enthalpy of the deformed metallic glass in the present work do show a remarkably high degree of rejuvenation that is an excellent starting point for further studies. We consider *Nature Communications* to be an appropriate journal, because the issues raised are relevant for glass science

generally, and not just for metallic glasses. For example, highly rejuvenated (chalcogenide) glasses are relevant in the development of phase-change computer memory. We are confident that the work will attract a wide attention.

Comments by Reviewer #3

We appreciate the positive comments by the reviewer, and focus on the concerns raised:

[1] Page 1, line 22 “We explore the effects of this rejuvenation, possibly near the achievable limit, on mechanical properties.” In this work, the degree of rejuvenation is defined by the energy increase or the hardness decrease from the as-cast state. Therefore, the degree of rejuvenation depends on the initial state. Is it possible to discuss the achievable limit of the rejuvenation based on this definition? If we prepare a well-aged glass and take it as an initial state, the upper limit of the rejuvenation becomes higher when we apply the present definition.

We take ‘rejuvenation’ to mean changing the glassy state to be characteristic of a higher cooling rate, and this is most directly revealed in terms of enthalpy, measured as the enthalpy of relaxation ΔH_{rel} . The extent of rejuvenation can then be considered in terms of either the increase in ΔH_{rel} , or the absolute value of ΔH_{rel} that is reached. In the manuscript we are careful to describe which of these we mean in each case. The reviewer is of course correct that a greater change in ΔH_{rel} can be achieved if the starting point is a well-aged glass. However, when considering limits, the absolute ΔH_{rel} reached is of more interest than the increment. The limit of rejuvenation achievable by mechanical deformation might be characterized, for example, by the lowest hardness and or the highest ΔH_{rel} . This limit would not depend on the initial state of the glass, whether as-cast or fully annealed.

[2] Page 4, line 147 “the enthalpy excess of the deformed glass over the relaxed glass is essentially the same as that of the relaxed glass over the state to which it would crystallize.” Does “the relaxed glass” mean the as-cast glass? As indicated in the previous comment [1], excess enthalpy of the deformed glass over the relaxed glass depends on the energy state of “the relaxed glass”. Therefore, there is a possibility that this sentence does not work well.

This point is most easily considered in terms of the practicalities of differential scanning calorimetry (DSC). In the present work a standard heating rate of 20 K min^{-1} is used. The enthalpy excess of the deformed glass is measured as the enthalpy of relaxation ΔH_{rel} . The reference ‘relaxed’ state is then the state reached after the release of ΔH_{rel} . Heats of crystallization measured at this rate give the enthalpy difference between the crystalline state of the sample and the same relaxed state (i.e. after release ΔH_{rel}). Thus, in effect, “relaxed glass” means the state reached after release of ΔH_{rel} on heating at 20 K min^{-1} . We have expanded the wording of this paragraph in order to clarify the point:

“The maximum ΔH_{rel} in the present work, 3.42 kJ mol^{-1} , is 66% higher than the highest previously reported, which was in a specimen subjected to 50 revolutions (maximum strain ≈ 37) in high-pressure torsion²⁵. This maximum enthalpy excess of the deformed glass over the relaxed glass (i.e. the glass after release of ΔH_{rel}) is 41% of the enthalpy of melting. For comparison, the heats of crystallization of metallic glasses are $\sim 40\%$ of the enthalpy of melting¹. Thus the difference in enthalpy between possible glassy states is essentially the same as that between the relaxed glass and the state to which it would crystallize.”

[3] Page 4, line 150 “Scaling calorimetric and viscosity data typical for a Zr-based BMG1,3,22,23, the highest energy in the present work corresponds to cooling at nearly 10^{10} K/s (Fig. 3c).” How do you estimate the cooling rate of 10^{10} K/s from the energy? In other words, how do you scale calorimetric measurement and viscosity data. Since “ 10^{10} K/s ” is one of keywords in this manuscript, it is better to provide a further explanation about this estimation in the “Methods” or in the supplementary information.

We are pleased to provide further information on this point in the revised manuscript. The cooling rate required to achieve an equivalent glassy state is certainly important in characterizing the degree of rejuvenation for a metallic glass. We hope that the added text (including two new references [26,27]) covers the point:

“The enthalpy of an as-cast glass is dependent on the cooling rate at which it was formed. The glass transition occurs at critical value of liquid viscosity η that is inversely proportional to the cooling rate. At the standard cooling rate of 20 K min^{-1} , the critical η is taken to be 10^{12} Pa s . From the measured

temperature dependence of η for Zr-based glass-forming liquids [Ref.26], the effective value of the glass-transition temperature T_g can then be estimated for other cooling rates. Given the dependence of T_g on cooling rate, the excess enthalpy (relative to that of a glass formed at the standard cooling rate) can be estimated from the temperature dependence of the liquid enthalpy [Ref.27], as shown in Fig. 1 of Ref. [2]. In this way, the excess enthalpy (ΔH_{rel}) of a glass can be related to the cooling rate at which a glass of that enthalpy would have formed from the liquid without further processing.”

[4] Page 5, line 170

“That these values are similar to those expected for the same strain applied at RT suggests that the rejuvenation is reaching saturation.” In the case of 77 K, the maximum strain is 33%, which is smaller than 40 % of RT. Thus, there still remains a possibility that the degree of rejuvenation becomes higher when we applied a larger strain than 33% at 77K. Compressive loading with a strain larger than 40 % at RT or than 33% at 77K should demonstrate the saturation more clearly and directly. The saturation is an important point of this work.

The reviewer is correct that the maximum applied strains are different at RT and at 77 K. However, interpolation does permit quantitative comparison. We consider three quantities: (i) the hardness in the centre of the specimen, (ii) the hardness at the notch root, and (ii) the heat of relaxation in the material at the notch root. In each case, the property varies roughly linearly with the imposed plastic strain; accordingly a linear interpolation can be used to estimate property values at intermediate strains, as shown in the Table:

Quantity	Values after deformation at RT to the nearest strains to 33%		Interpolated value at 33% strain at RT	Measured value at 33% strain at 77 K
Hv at centre of specimen	425±8 at 30%	401±14 at 40%	418±10	440±13
Hv at notch root	396±16 at 20%	315±16 at 40%	343±16	335±17
ΔH_{rel} at notch root (kJ/mol)	0.49±0.04 at 0%	3.42±0.08 at 40%	2.91±0.08	3.29±0.08

Comparison of the measured and interpolated values, and noting the error bars,

suggests that the effects of deformation are indeed similar at RT and 77 K. As we point out in the revised manuscript, the lack of importance of temperature can be understood in terms of the rates of generation and annihilation of free volume. Under compression at RT, the annihilation rate is already effectively zero, so being colder makes no difference.

Our earlier study (Pan et al. *Acta Mater.* **59** (2011) 5146–5158) on softening of a single shear band showed that the minimum hardness of within single shear band did not change regardless of the overall plastic strain (2–6%) and the maximum reduction in hardness was constant at 36%. This is a clear sign of saturation, but it is for the operating conditions in a single shear band, which may be different from what happens in the notched sample. The constraint in the notched samples might be too high, so the strain is not strong enough to reach the saturation under our current conditions.

So we tend to agree with the reviewer and have removed most of the mentions of saturation. In the future, we believe that we could impose less constraint, allowing more shear (and therefore more free volume generation) at lower compressive pressure (but still enough to suppress annihilation). Of course, the constraint must still be strong enough to suppress shear banding. It seems that we might already be achieving this (lesser constraint) at the notch root, but there is no reason to suppose that the conditions there are optimal — so the degree of rejuvenation might not be at saturation. To properly address the reviewer’s point about saturation will require measurements on different sample geometries and strains— and that is beyond the scope of the present work.

REVIEWERS' COMMENTS:

Reviewer #1 (Remarks to the Author):

The revised manuscript is substantially improved by the additional work. In particular the new Fig. 1 (c) is very convincing and revealing. I suspect this map is very different for the case of tension, presumably much more localized around the notch. I am afraid the explanation of the difference between tension and compression in terms of the activation volume is too simplistic. I guess the difference in the deformed volume and the degree of stress concentration must also be considered. However, that can be delegated to the next publication. I recommend accepting the manuscript as revised.

Reviewer #2 (Remarks to the Author):

In the revised version of the manuscript, the authors now do cite their earlier work. We remind ourselves that the earlier work published in PRL is practically identical, with the difference that the authors have done the test in tension. In the current manuscript, the exact same methodology is followed, but now the samples is loaded in tension. And, as the authors write in their response, they expected that rejuvenation would occur in compression. This is indeed stated in the original work published in PRL.

Given this, what is this new manuscript submitted to NatCom actually contributing?

Let us look at (some of) the literature on energy storage in metallic glasses:

Acta Materialia 56 (2008) 5440-5450: elastostatic rejuvenation via compressive loading

Applied Physics Letters 94 (2009) 021907: elastostatic rejuvenation via compressive loading

Scripta Materialia 64 (2011) 966: dilatation (and thus rejuvenation) induced by compressive loading

Physical Review Letters 111 (2013) 135504: densification and relaxation in tension

Scripta Materialia 102 (2015) 67: relaxation induced by compressive loading

Philosophical Magazine 96 (2016) 1643: energy storage via tensile creep

Acta Materialia 138 (2017) 111: rejuvenation via cyclic elastic loading

Applied Physics Letters 110 (2017) 111901: rejuvenation under hydrostatic pressure

Journal of Applied Physics 121 (2017) 205109: densification and rejuvenation under hydrostatic pressure

Scientific Reports 7 (2017) 625: rejuvenation and relaxation obtained in compressive loading, identification of a critical stress needed to produce either or

With the above list of reports it is clear that energy storage via mechanical loading (various stress states) is nothing new. I am really not sure what the current manuscript adds to what we already have learned. I would argue that a new insightful study would actually clear the discrepancies and contradictions of the foregoing suite of papers.

Again, the presented manuscript is a fine piece of work, but since it basically is a copy of the same authors earlier work, with the only difference being compression instead of tension, I think the current work is rather a case for Scientific Reports.

Reviewer #3 (Remarks to the Author):

The authors have considered my concerns and comments and added revisions to the manuscript. In the revised manuscript, results and discussion are supported by both various experiments and sufficient explanation. I would recommend the publication of the revised manuscript in Nature Communications.

We thank all the reviewers for their careful and constructive comments on the revised version of our manuscript. Our responses to these comments are as follows:

Comments by Reviewer #1

Reviewer #1 (Remarks to the Author):

The revised manuscript is substantially improved by the additional work. In particular the new Fig. 1 (c) is very convincing and revealing. I suspect this map is very different for the case of tension, presumably much more localized around the notch. I am afraid the explanation of the difference between tension and compression in terms of the activation volume is too simplistic. I guess the difference in the deformed volume and the degree of stress concentration must also be considered. However, that can be delegated to the next publication. I recommend accepting the manuscript as revised.

We agree with the reviewer that the hardness map for the case of tension may be different from that for compression. Indeed, we had already started work on a map for the case of tension before receiving the latest set of comments. More broadly, we agree that the cases of tension and compression are not simply 'equal and opposite'. Nevertheless, we do maintain that our results provide clear evidence for the importance of the hydrostatic component of the applied stress state. We note that, on the basis of the present results, the reviewer recommends acceptance of the revised manuscript.

Comments by Reviewer #2

Reviewer #2 (Remarks to the Author):

In the revised version of the manuscript, the authors now do cite their earlier work. We remind ourselves that the earlier work published in PRL is practically identical, with the difference that the authors have done the test in tension. In the current manuscript, the exact same methodology is followed, but now the samples is loaded in tension. And, as the authors write in their response, they expected that rejuvenation would occur in compression. This is indeed stated in the original work published in PRL.

Given this, what is this new manuscript submitted to NatCom actually contributing?

Let us look at (some of) the literature on energy storage in metallic glasses:

Acta Materialia 56 (2008) 5440-5450: elastostatic rejuvenation via compressive loading

Applied Physics Letters 94 (2009) 021907: elastostatic rejuvenation via compressive loading

Scripta Materialia 64 (2011) 966: dilatation (and thus rejuvenation) induced by compressive loading

Physical Review Letters 111 (2013) 135504: densification and relaxation in tension

Scripta Materialia 102 (2015) 67: relaxation induced by compressive loading

Philosophical Magazine 96 (2016) 1643: energy storage via tensile creep

Acta Materialia 138 (2017) 111: rejuvenation via cyclic elastic loading

Applied Physics Letters 110 (2017) 111901: rejuvenation under hydrostatic pressure

Journal of Applied Physics 121 (2017) 205109: densification and rejuvenation under hydrostatic pressure

Scientific Reports 7 (2017) 625: rejuvenation and relaxation obtained in compressive loading, identification of a critical stress needed to produce either or
With the above list of reports it is clear that energy storage via mechanical loading (various stress states) is nothing new. I am really not sure what the current manuscript adds to what we already have learned. I would argue that a new insightful study would actually clear the discrepancies and contradictions of the foregoing suite of papers.
Again, the presented manuscript is a fine piece of work, but since it basically is a copy of the same authors earlier work, with the only difference being compression instead of tension, I think the current work is rather a case for Scientific Reports.

We are pleased that the reviewer now considers this to be “a fine piece of work”, and recognises that it is consistent with (rather than contradicting) the earlier results in tension (Wang et al., *PRL* **111** (2013) 135504. The reviewer cites several earlier studies, notes that “energy storage via mechanical loading is nothing new”, asks “what is this new manuscript ... actually contributing?” and seeks “a new insightful study [that] would actually clear the discrepancies ... of the foregoing ... papers”.

We certainly agree that, by now, there has been much work on energy storage (rejuvenation) by mechanical loading. In our view, the present manuscript makes significant progress in understanding the mechanism of rejuvenation and in resolving discrepancies in earlier results. It is novel in demonstrating a dramatically greater degree of rejuvenation than has been achieved before.

The reviewer argues that the manuscript is “basically a copy of the same authors earlier work, with the only difference being compression instead of tension”. We argue, however, that the close similarity of the earlier experiments (in tension) and the present experiments (in compression) is particularly useful: that they give opposite results (respectively relaxation and rejuvenation) assists greatly in understanding the role of the hydrostatic stress component in mechanical loading.

In our view, the manuscript makes significant contributions that are new:

- (i) Mechanical loading in the elastic regime is efficient in achieving rejuvenation, yet the extent of rejuvenation is small, because the strains are so low. On the other hand, plastic deformation, achieving large strains, is inefficient in storing energy. This inefficiency is associated with the localization of the deformation in shear bands (for reasons set out in the manuscript). The key contribution of the present work is to show that loading of a notched specimen can achieve significant plastic strains without any evident shear banding.
- (ii) In compressively loaded notched samples, the hardness decreases and the heat of relaxation increases. The nature of these changes, compared to those induced by other mechanical treatments, suggests that a new regime of homogeneous plastic flow, at room temperature and below, has been achieved. In this regime, nearly 30% of the mechanical work done is stored as increased enthalpy in the metallic glass. This efficiency of energy storage is roughly one order of magnitude higher than any reported before.

- (iii) Most of the literature on plastic deformation on metallic glasses considers effects as a function of shape change (characterized by shear strain), with little attention to the effect of the hydrostatic component of the applied stress. It is accepted that the flow stress of metallic glasses is dependent on the hydrostatic component of stress, but that dependence is weak. Based on our directly comparable studies in tension and compression, the results in this paper show unambiguously (in our view for the first time) that the hydrostatic component of the applied stress has a very important role in understanding the effects of deformation. It is striking that (for similar shear strain) the effects of deformation can be opposite (relaxation vs rejuvenation) depending on the hydrostatic component of the applied stress. The new studies of the effect of deformation temperature help to confirm the role of underlying ‘damage’ and ‘repair’ rates (for which one possible description is generation and annihilation of free volume) in governing the extent of rejuvenation or relaxation.
- (iv) The constrained loading geometry achieves property changes in ‘bulk’ (i.e. not confined to near the central plane of a shear band) metallic glass that greatly exceed any previously reported results of deformation. Characterised by the increase in enthalpy, the demonstrated effects are 66% greater than previously realised by any plastic deformation, and roughly four times greater than any achieved by mechanical treatments in the elastic regime.
- (v) The most extreme state achieved by in the present work has an energy comparable to that expected on quenching the liquid at 10^{10} K s⁻¹. This is simply unprecedented for mechanically induced changes in a bulk glass, and yet it may not be the limit. Mapping of the extent of rejuvenation on a cross-section through a notched sample reveals a complex pattern. Further analysis of this pattern should assist in optimizing methods for bulk rejuvenation.

These points were already made in the manuscript, but we have given them some more prominence in the concluding paragraph in the main text. (Added text is indicated in red.)

Comments by Reviewer #3

Reviewer #3 (Remarks to the Author):

The authors have considered my concerns and comments and added revisions to the manuscript. In the revised manuscript, results and discussion are supported by both various experiments and sufficient explanation. I would recommend the publication of the revised manuscript in Nature Communications.

We are pleased that the reviewer recognises our revisions to the manuscript and now recommends acceptance without further revision.